# Effect of the Lys62Ala Mutation on the Thermal Stability of *Bst*HPr Protein by Molecular Dynamics

**DOI:** 10.3390/ijms25126316

**Published:** 2024-06-07

**Authors:** Aranza C. Martínez-Zacarias, Edgar López-Pérez, Salomón J. Alas-Guardado

**Affiliations:** 1Departamento de Ciencias Naturales, Universidad Autónoma Metropolitana Unidad Cuajimalpa, Ciudad de México 05300, Mexico; arafja12@gmail.com; 2Posgrado en Ciencias Naturales e Ingeniería, Universidad Autónoma Metropolitana Unidad Cuajimalpa, Ciudad de México 05300, Mexico; edgarlopezperez07@gmail.com

**Keywords:** *Bst*HPr protein, wild type, mutant, thermal stability, molecular dynamics, salt bridge network, molecular staple

## Abstract

We analyzed the thermal stability of the *Bst*HPr protein through the site-directed point mutation Lys62 replaced by Ala residue using molecular dynamics simulations at five different temperatures: 298, 333, 362, 400, and 450 K, for periods of 1 μs and in triplicate. The results from the mutant thermophilic *Bst*HPrm protein were compared with those of the wild-type thermophilic *Bst*HPr protein and the mesophilic *Bs*HPr protein. Structural and molecular interaction analyses show that proteins lose stability as temperature increases. Mutant and wild-type proteins behave similarly up to 362 K. However, at 400 K the mutant protein shows greater structural instability, losing more buried hydrogen bonds and exposing more of its non-polar residues to the solvent. Therefore, in this study, we confirmed that the salt bridge network of the Glu3–Lys62–Glu36 triad, made up of the Glu3–Lys62 and Glu36–Lys62 ion pairs, provides thermal stability to the thermophilic *Bst*HPr protein.

## 1. Introduction

Proteins are macromolecules made up of chains of amino acids and play an important role in the structure and function of cells. They have numerous and important functions in the organism, as they can carry out different processes such as enzymatic catalysis, molecular transport, mechanical resistance, protection against pathogens, metabolic regulation, etc. [1,2]. For a protein to perform these functions, it must maintain a stable fold, which is achieved during the transcription of the mRNA information in the ribosome. The chemical nature of the amino acids and the environment in which the proteins are found are essential for suitable folding [3].

A large number of proteins have been characterized thanks to advances in the fields of molecular biology and biochemistry. Structural information and thermodynamic data have been the basis for the development of algorithms and methods for the explanation of protein-folding mechanisms [1,2]. In addition, thermodynamic data of these biomolecules are essential to understanding and predicting the stability when mutations are made to improve their resistance to unfolding under the influence of external factors such as solvents, salts, temperature, etc. [4,5,6].

The study of proteins from extremophilic organisms has generated great interest in the field of protein folding, since the elucidation of this process contributes to better knowledge of the mechanisms that stabilize proteins, which is fundamental to improving their design, for example by developing more efficient enzymes that work at high temperatures [7,8,9].

Proteins from thermophilic and hyperthermophilic organisms show stable adaptations to high-temperature environments, e.g., hydrothermal vents, hot springs, mud volcanoes, etc. [10]. In recent decades, studies have focused on analyzing the unusually high stability of these proteins [4,5,6,7,8,9]. The knowledge gained has been used to improve proteins from other types of organisms, such as mesophiles. There are different reasons for this, the most common being that temperature is a variable used in most industrial processes, for example, in the food industry [7]. Research has suggested that there are several mechanisms for increasing the thermal stability of proteins. However, none of them provide a universal solution, as it has been found that proteins activate different physicochemical pathways to withstand high temperatures and not lose their functionality within the ecosystems where their host organisms live [10]. An alternative way to study and improve the thermal stability of proteins is to perform point mutations of sites, i.e., mutations can be performed by making a targeted exchange of amino acids [6,11,12]. Experimental studies such as circular dichroism (CD), differential scanning calorimetry (DSC), and fluorescence spectroscopy are used to determine whether thermal resistance increases [5,13].

A particular example of an extremophilic organism is *Bacillus stearothermophilus* (*Bst*), which is considered thermophilic [12,13,14,15,16] or can be classified as moderately thermophilic [17]. As with all organisms, this microorganism contains macromolecules involved in metabolic processes, including the histidine-containing phosphocarrier (HPr) protein, which is a key factor in the bacterial phosphoenolpyruvate:sugar phosphotransferase (PTS) system [12,16,18,19].

The thermostability analyses of the *Bst*HPr protein have been reported in various experimental studies [12,13,14,15,16]. Their structural and thermodynamic data have been compared with those of homologous proteins from mesophilic organisms such as *B. subtilis* (*Bs*), *E. coli* (*Ec*), *E. faecalis* (*Ef*), and *M. capricolum* (*Mc*); moderate thermophiles such *S*. *thermophilus* (*St*); thermophiles as *T. tengcongensis* (*Tt*); and haloalkaliphilic organisms such as *B. halodurans* (*Bh*) and *O. iheyensis* (*Oi*) [12,14,16]. Stability measurements of the *Bst*, *Bh*, *St*, *Bs*, and *Oi* HPr proteins carried out by thermal and solvent denaturation, under different experimental conditions of pH, salinity, and temperature, have shown that *Bst*HPr has the highest thermal stability. For example, Table 1 shows a comparison of thermodynamic data, such as free energy of stabilization (∆G_S_), temperature of maximal stability (T_S_), melting temperature (T_m_), change in heat capacity (∆C_p_), and change in enthalpy (∆H), between the *Bst*HPr and *Bs*HPr proteins at pH = 7.0 [12,13,14].

In addition, it has been observed that the *Bst*HPr protein forms a salt bridge between the Asp11 and Lys57 residues [12,16]. The site-directed Lys57Thr mutation shows that the Asp11-Lys57 salt bridge plays an important role in the thermostability of the *Bst*HPr protein [16]. These facts have shed light on the structural behavior and the mechanisms involved in the thermal stability of the *Bst*HPr protein but have not been conclusive, since the key molecular interactions that stabilize it at elevated temperatures are still not well understood.

Recently, our group performed molecular dynamics (MD) simulations of HPr proteins from *Bacillus stearothermophilus* (*Bst*HPr) and *Bacillus subtilis* (*Bs*HPr) organisms to elucidate the molecular mechanisms that provide thermal resistance to the thermophilic protein at elevated temperatures. Structural and molecular interaction results showed that the salt bridge network formed by the Glu3–Lys62–Glu36 triad is a key factor in its stability [20]. To confirm this hypothesis, in this work, we present MD analysis of a mutant structure (*Bst*HPrm) consisting of the replacement of the lysine residue at position 62 with an alanine residue, affecting the aforementioned triad. Moreover, the effects of temperature on the mutant structure of the HPr protein were analyzed and compared with those obtained from the thermophilic wild-type (*Bst*HPr) and mesophilic (*Bs*HPr) structures.

## 2. Results

### 2.1. Structural Behavior

Figure 1 shows the root mean square deviation (RMSD), radius of gyration (Rg), and fraction of native contact (Q) trajectories for *Bst*HPr and *Bst*HPrm proteins at 298, 362, and 400 K. Both proteins exhibit almost the same structural behavior during the simulation time at 298 K (Figure 1a,d,g), indicating that their structures are stable. However, when the temperature increases at 362 K, the mutant protein shows slight structural increases in its overall fluctuations, expansions, and loss of native contacts, for example, in the range of 300 to 400 ns, compared with the wild-type protein (Figure 1b,e,h). These changes become more pronounced as the temperature increases to 400 K. It is observed that after 100 ns the mutant protein displays larger global fluctuations, undergoing drastic structural expansion and compaction states with significant losses of native contacts (Figure 1c,f,i). In particular, around 490 ns the *Bst*HPrm protein shows larger fluctuation and structural expansion than the *Bst*HPr protein.

Figure 2 shows the average values (avg) and their corresponding standard deviation (SD) of the RMSD (Figure 2a), Rg (Figure 2b), and Q (Figure 2c) of the *Bs*HPr, *Bst*HPr, and *Bst*HPrm proteins. It should be noted that the *Bs*HPr tag refers to the HPr protein from the mesophilic organism *Bacillus subtilis*, whose results are reported in reference [20]. In this figure, it is observed that the mutant protein presents almost the same structural behavior as the mesophilic protein. In particular, (a) the average values of RMSD and Q are practically identical at a temperature higher than 362 K and (b) the mutant protein remains more compact than the mesophilic one in the unfolded state at 450 K, as measured by the Rg.

Figure 3 shows the behavior of the β-strand and α-helix native secondary structures (SS) of the three proteins. These proteins contain the same number of secondary structures at 300 K, i.e., the content of β-strand and α-helix is approximately 26 and 34%, respectively; thus, around 40% of the secondary structure is random coil. Moreover, it is observed that with increasing temperature, the *Bs*HPr and *Bst*HPrm proteins lose β-strand and α-helix structures compared to the *Bst*HPr protein. The most affected structures are the α-helices since they show higher losses at lower temperatures than the β-strands, i.e., more α-helix structures are lost from 362 K, while the β-strands lose a higher structural percentage from 400 K onwards. Additionally, the average percentage of β-strands lost from the mutant protein is between those of the wild-type and mesophilic proteins at 400 and 450 K (Figure 3a). On the other hand, the average percentage of the α-helices lost by the mutant protein is almost identical to that of the mesophilic protein at 362 and 400 K (Figure 3b), ensuring that α-helix structures are more affected than the β-strand ones as the temperature increases.

Considering only the β-strands of the mutant protein, the β_4_-strand shows greater structural loss, which is to be expected since the mutation has been made in this structure. As expected, the other structures affected are the β_1_- and β_2_-strands, which contribute with their Glu3 and Glu36 residues to form the Glu3–Lys62–Glu36 salt bridge network. Therefore, the β_3_-strand does not present structural modifications, i.e., the structural loss is similar to that of the wild-type protein.

On the other hand, considering the α-helices of the same protein, the mutation of site 62 causes the α_1_-helix to be the most affected of these structures, while the α_2_-helix remains almost unchanged to that of the wild-type protein. The changes produced in the α_1_-helix may occur because the Glu84–Arg17 salt bridge decreases its frequency of formation, i.e., the mutation causes the weakening of the salt bridge; thus, the secondary structure is destabilized. Details of these facts will be discussed later. The individual behavior of each β-strand and α-helix is provided in Appendix A, while the secondary structure profiles are given in Appendix A. Those profiles were calculated using the “define secondary structures of proteins” (DSSP) algorithm [20].

### 2.2. Molecular Interactions

We first analyzed the hydrogen bonds (HBs) in both proteins, which are classified into two types: (1) bonds that form between residues of the secondary structures (labeled HBpp) and (2) interactions between protein residues with the solvent (labeled HBps).

Figure 4 shows the trajectories of the HBpp and HBps values of the *Bst*HPr and *Bst*HPrm proteins at the temperatures of 298, 362, and 400 K. Both proteins show similar trends for the two types of hydrogen bonds. Only slight changes in these trends are observed; for example, HBpp decreases in the *Bst*HPrm protein between 100 and 500 ns at 400 K (Figure 4c) and HBps decreases in the *Bst*HPr protein at around 150–500 ns at 400 K (Figure 4f).

Figure 5 shows the average values and standard deviations of both hydrogen bonds for the *Bs*HPr, *Bst*HPr, and *Bst*HPrm proteins at the five temperatures studied. The HBpp trends of the three proteins are similar, while the HBps trend is similar between *Bst*HPr and *Bst*HPrm proteins. Yet, these values in the mesophilic protein are higher at all temperatures; this occurs because the *Bs*HPr protein possesses a higher number of polar residues that are exposed to the solvent [20].

Taking as reference the average values of the HBpp and HBps of the *Bst*HPr and *Bst*HPrm protein structures at 298 K, it is possible to calculate the number and percentage of hydrogen bonds lost for each temperature, which is summarized in Table 2 and Table 3.

These analyses show that both proteins lose approximately the same number and percentage of HBpp and HBps from 298 to 450 K. However, the mutant protein loses more HBpp at 400 K (19.5%), indicating that the interior of its structure is weakened more than that of the wild-type protein at this point.

Secondly, we analyzed the solvent accessible surface area (SASA) behavior of both proteins for two residue types: polar (labeled SASAp) and non-polar (labeled SASAnp). Figure 6 shows the trajectories of the SASAp and SASAnp values of the *Bst*HPr and *Bst*HPrm proteins at 298, 362, and 400 K. From this figure, it can be seen that both trends increase for the two protein structures as the temperature increases. However, the SASAp trend of the *Bst*HPrm protein is higher than that of the *Bst*HPr one, indicating that the mutation causes greater exposure of the polar residues to the solvent. While the SASAnp of the *Bst*HPrm protein is slightly lower than that of the *Bst*HPr protein at 298 K, this behavior changes as the temperature increases, i.e., the SASAnp of the *Bst*HPrm protein is higher compared to that of the *Bst*HPr protein at 400 K. Therefore, at low temperatures (e.g., 298 K) the non-polar residues of the mutant protein are more buried inside its structure; however, these lose stability as the temperature increases (e.g., 400 K), causing greater contact with the solvent in comparison to the wild-type protein.

Considering the average SASAp and SASAnp values of the *Bst*HPr and *Bst*HPrm proteins, one can calculate the amount and percentage of area that increased or decreased in the thermophilic protein due to the mutation, which is shown in Table 4.

The surface area of polar residues increased for all temperatures. It should be noted that the largest increase is observed at 400 K, whereas the surface area of the non-polar residues increases only at 400 K (values denoted with an asterisk), but this value decreases for the other temperatures.

These analyses confirm that the polar residues of the *Bst*HPrm protein are more exposed to the solvent than those of the *Bst*HPr protein. Furthermore, the non-polar residues of the *Bst*HPrm protein are more buried inside the structure at low temperatures, e.g., at 298 K (see Figure 6d), but these residues become more exposed to the solvent as the temperature increases, e.g., at 400 K (see Figure 6f).

Figure 7 shows the average values and their corresponding standard deviations of the SASAp and SASAnp of the *Bs*HPr, *Bst*HPr, and *Bst*HPrm proteins at the five temperatures analyzed. The SASAp behaviors of the *Bs*HPr and *Bst*HPrm proteins are similar, while their SASAnp trends are lower than the values of the *Bst*HPr protein at 298 and 333 K. However, as stated in the previous paragraph, from 362 K onwards it is observed that the mutant protein changes its tendency concerning the wild-type protein. In addition, the same behavior is observed in the mesophilic protein.

In addition to calculating the behavior of the hydrophobic core residues through SASA, we performed the analysis of the ILV clusters of the mutant protein at 750 ns at the different temperatures studied. The ILV clusters for both proteins at 298, 362, and 400 K are given in Figure 8. These results show how these clusters change with the temperature. In particular, the mutant protein forms three clusters at 400 K, while the thermophilic protein forms one cluster at the same temperature. This fact provides insight into the stability of the hydrophobic cluster of the *Bst*HPr protein because the point mutation Lys62Ala triggers structural instability, causing the hydrophobic core to be exposed to the solvent, as previously described with the SASA analyses.

Additionally, we analyzed the formation of ionic pairs in both proteins. In our previous work [20], in which the thermal stability of the *Bst*HPr protein was analyzed, the formation of five salt bridges (SBs) was observed: Asp79–Lys83, Glu84–Arg17, Asp11–Lys57, Glu3–Lys62, and Glu36–Lys62, where the latter two ionic pairs form the Glu3–Lys62–Glu36 triad. The residues of this triad of Glu3, Lys62, and Glu36 are located on the β_1_-, β_4_-, and β_2_-strands, respectively. Thus, the structural and physicochemical nature of the β_4_-strand is expected to be affected, since the Lys62 residue has been mutated by Ala62. Except for the Asp79–Lys83 salt bridge, which is of the intra-molecular type, the other SBs are formed between residues that are in different secondary structures, i.e., they are of the inter-molecular type. Table 5 shows these values.

On the other hand, in this work, in which we analyzed the thermal stability of the *Bst*HPr protein but performed the point mutation Lys62Ala, the formation of four salt bridges was observed in the *Bst*HPrm protein: Asp79–Lys83, Glu84–Arg17, Asp11–Lys57, and Glu32–Lys45. The SB Glu32–Lys45 is the only new ionic interaction formed in the mutant protein. Table 6 shows the average frequency of each SB from the three simulations of the *Bst*HPrm protein for the temperatures studied.

Considering the values in both Table 5 and Table 6, it can be seen that the mutation causes the following changes in these molecular interactions with increasing temperature:(a)The Asp79–Lys83 salt bridge, located on the α_3_-helix, does not undergo meaningful changes; only its frequency is slightly higher at 333 K.(b)The Glu84–Arg17 salt bridge, formed between the α_3_-helix and α_1_-helix, undergoes significant changes, since it presents lower frequencies at all temperatures, causing the secondary structures to be less stable.(c)The Asp11–Lys57 salt bridge, formed between two loops (that are in the β_1_-strand/α_1_-helix and the α_2_-helix/β_4_-strand structures), increases its frequency at 298 and 333 K but decreases for the other temperatures. In particular, it decreases drastically at 400 K.(d)The Glu32–Lys45 salt bridge is formed between the β_2_-strand and the first residue of the loop after the β_3_-strand. This interaction is very weak, as its frequency is less than 0.3 at 298 and 333 K. It reaches values slightly higher than 0.3 at 362 and 400 K, but it is almost lost at 450 K.

In addition to these frequency measurements, we calculated the distances of the salt bridges for both proteins. These values can be found in Appendix A. The structural comparison between the residues of the Glu3–Lys62–Glu36 triad in *Bst*HPr and the corresponding residues of the *Bst*HPrm and *Bs*HPr proteins are given in Appendix A.

A salient feature to highlight is that due to the mutation, different regions of the protein lose structural strength. For example, the SB Glu84–Arg17 is weakened, as it presents lower frequencies compared to those observed in the wild-type *Bst*HPr protein, indicating that the Lys62 residue forming the Glu3–Lys62–Glu36 triad is crucial in the structural stability of the protein.

Lastly, we performed a brief analysis of the flexibility of the active site in both proteins. The main function of the HPr proteins is to transfer a phosphate group within the PTS system in bacteria [12,16,19]. The flexibility of two residues belonging to the active center of the HPr protein, His15 and Arg17, is essential for this transport. Depending on the spatial relationship between these residues, the HPr protein can adopt two main conformations: open state (OS) and closed state (CS). The OS is considered when the distance between the N^δ1^ of His15 and N of Arg17 is *d*_Nδ1-N_ > 7.5 Å, and the CS when this distance is *d*_Nδ1-N_ < 4.5 Å. It has been proposed that dynamic conversion between the OS and the CS is essential for the phosphorylation of this protein and its effect as a transcriptional regulator [19]. Through the simulations performed in this study, it was possible to observe that *Bst*HPr and *Bst*HPrm proteins adopt both states frequently during the MD trajectory; hence, the ratio of OS/CS was calculated according to the total number of states achieved, as indicated in Figure 9.

## 3. Discussion

*Bst*HPr and BstHPrm proteins lose their structural hierarchy with increasing temperature, as the dominant arrangements at high temperatures no longer correspond to α-helices and β-strands but rather to less ordered structures such as loops and turns. The α-helix and β-strand secondary structures are stable at 298 K and almost all are lost at 450 K (Figure 3), indicating that both proteins are unfolded. However, it can be claimed that the most structurally stable protein throughout the simulations is the wild-type *Bst*HPr variant. This conclusion has been reached because, when comparing the secondary structures of the two proteins, the β_4_-, β_1_-, and β_2_-strands and the α_1_- and α_3_-helices in the *Bst*HPrm protein are more destabilized up to 400 K.

RMSD analysis shows that both proteins maintain their structural stability and almost similar behavior between 298 and 362 K (Figure 1a,b and Figure 2a), which agrees with the secondary structure profiles. At the temperature of 400 K, there is a difference in fluctuations between the proteins, with the *Bst*HPrm protein exhibiting larger structural global fluctuations than the *Bst*HPr one (Figure 1c). At the temperature of 450 K, drastic fluctuations are observed in the trajectories of the three simulations for both proteins (refer to Appendix A). This indicates a noticeable structural difference compared to the reference or native structures at 298 K, showing that both proteins have reached the unfolded state.

The radius of gyration of both proteins increases its average value and standard deviation between 298 and 450 K (Figure 2b), indicating that the expansion and compaction processes of the protein structures are meaningful as the temperature increases. In particular, this fact is noticeable at 450 K, since at this point structural unfolding occurs (as shown in Appendix A). Furthermore, the mutant variant experiences greater structural contractions and expansions than the wild-type variant at 400 K (Figure 1f), which is consistent with the RMSD behavior and secondary structure conformations.

The fraction of native contacts also decreases in both proteins (Figure 2c) from 298 to 450 K, i.e., topological interactions are lost regarding the initial structures (conformations at *t* = 0 ns). This statement is consistent with the aforementioned analyses, specifically the analysis of secondary structures (Figure 3), which indicates that the proteins lose a large number of their α-helix and β-strand structures at 450 K. However, it is clear that out of the two proteins, and as expected, the native contacts of the *Bst*HPrm protein decay more than those of the *Bst*HPr protein, which is illustrated in Figure 1h,i and Figure 2c. In particular, these contacts fall more in the *Bst*HPrm protein at 400 K.

It is known that each hydrogen bond in proteins contributes energetically on average 1 kcal/mol [22]; therefore, increasing the number of HBs improves the thermostability of these biomolecules. However, in this study, HBpp and HBps interactions decreased for both proteins with increasing temperature (Figure 4 and Figure 5). This occurs because the energy yielded to the system causes an increase in the kinetic energy of the particles, promoting the rupture of HB. If the number of HB decreases, then secondary structures are drastically lost, global fluctuations and compaction/expansion states increase, and more native contacts are lost, as described in the previous paragraphs.

In this way, comparing the behavior of the two proteins, it is observed that the reductions in HBpp and HBps are almost similar up to 362 K. Nonetheless, when the temperature increases to 400 K, more HBpp of the *Bst*HPrm protein is lost than in the *Bst*HPr protein, e.g., in the range from 298 to 400 K 19.5 and 15.1% is lost, respectively, whereas between 362 and 400 K 14.0 and 9.2% is lost, respectively. The opposite case arises with HBps since it increases in the mutant variant and decreases in the wild-type protein, e.g., HBps decreases by 1.6% in the *Bst*HPr and increases by 0.5% in the *Bst*HPrm between 362 and 400 K. These facts occur because the mutant protein is destabilized and unfolds more at this temperature (400 K). The residues buried in the structure are exposed to the solvent and the HBpp interactions are broken; consequently, they tend to form more hydrogen bonds with the solvent. This agrees with the structural analyses of RMSD, Rg, Q, and SS profiles, since at 400 K the most meaningful changes between the two proteins are observed. Additionally, from the native to the unfolded state (298–450 K), HBpp decreases by 33.3 and 33.9% for *Bst*HPr and *Bst*HPrm proteins, respectively, and HBps shows losses of 6.6% for *Bst*HPr and 7.0% for *Bst*HPrm, i.e., both proteins lose almost the same number of HBs, yielding structural similarities at 450 K.

Hydrophobic interactions and packing of the hydrophobic core of proteins are calculated indirectly through SASA. This is achieved by calculating the area of the proteins in contact with the solvent. The measurement is made on the total or part of the residues exposed to the solvent and can be distributed into polar and non-polar residue contacts. SASA is another useful parameter that indicates the loss of stability and unfolding of proteins, since hydrophobic zones are exposed when increasing the temperature, which usually interact with each other through non-polar residues that hardly have contact with the solvent. Consequently, in this work, the hydrophobic zones remain stable in both *Bst*HPr and *Bst*HPrm proteins at 298 K; nevertheless, as the temperature increases up to 450 K, these zones are exposed to the medium, i.e., the interaction between the non-polar residues decreases, leading to the instability of the hydrophobic core. The hydrophobic core reaches its maximum exposure for the two proteins at 450 K (Figure 7).

The Lys62Ala mutation causes the polar area to increase by an average of 8.7% at 298 K, keeping this trend almost constant up to 450 K, where the average difference is equal to 5.9%; that is, although the temperature increases, the difference in polar area between the proteins does not change significantly. Conversely, the mutation causes the non-polar area to decrease slightly up to 362 K; however, this area increases at 400 K by 2.3%, i.e., the hydrophobic core of the *Bst*HPrm protein is more exposed to the solvent than that of the *Bst*HPr protein at this temperature. In other words, the hydrophobic interactions of *Bst*HPrm are weaker at this point, losing more structural stability. In addition, SASAnp increases by an average of 14.3% for the *Bst*HPrm protein and by an average of 8.3% for the *Bst*HPr protein in the range of 298 to 400 K. The proteins reach their maximum non-polar area exposure at 450 K; as such, the area increases by 39.0 and 39.7% for *Bst*HPr and *Bst*HPrm, respectively, between 298 and 450 K. In addition to these results, the ILV cluster analysis shows that the hydrophobic core of the mutant protein loses stability at 400 K as the number of these clusters increases. These facts are consistent with the structural and hydrogen bond analyses, as discussed in the aforementioned paragraphs.

Salt bridges play an important role in the stability of proteins, especially when they are exposed to high temperatures. In this work, the mutant protein conserves the salt bridges Asp79–Lys83, Glu84–Arg17, and Asp11–Lys57 from the wild-type protein. The Asp79–Lys83 salt bridge is the most stable one to temperature changes during MD simulations since it does not show any significant variations in its average formation frequency compared to the *Bst*HPr protein, and according to our frequency criterion, its value is greater than 0.3 up to 450 K. The stability of this SB is due to its intramolecular nature, i.e., it is located on the α_3_-helix structure, which is slightly affected by temperature (see Appendix A). On the other hand, the Glu84–Arg17 salt bridge is the least stable to temperature changes, as its average formation frequency decreases largely upon mutation (Table 6). The mutation Lys62Ala causes the formation of the Glu32–Lys45 salt bridge in the *Bst*HPrm variant. This interaction is unlikely to contribute to its stability since the formation frequency is less than 0.3 at 298 K; that is, it presents an average frequency of 0.246. However, it may provide some structural stability when increasing to 362 and 400 K, since the average frequency increases to 0.334 and 0.351, respectively. The β_1_- and β_4_-strands are the secondary structures most affected by the mutation, as they lose more structure when the temperature reaches 400 K (see Appendix A). Additionally, more β-strand fragments are observed in the *Bst*HPr protein than in the *Bst*HPrm one at 450 K (see Appendix A), indicating that the mutation affects the structural behavior of these structures up to the unfolded state. Analyzing these facts and comparing them with the results of the parameters RMSD, Rg, Q, SS, HB, and SASA, it is gathered that the salt bridge network formed by the Glu3–Lys62–Glu36 triad provides greater thermal stability to the *Bst*HPr protein. These parameters are less affected in the wild-type protein than in the mutant protein up to 400 K.

An interesting fact is that when comparing the behavior of the structural (RMSD, Rg, Q, and SS) and molecular interaction (HB and SASA) parameters of the *Bst*HPr and *Bst*HPrm proteins with their mesophilic *Bs*HPr counterpart, the mutant protein shows identical trends to those of the mesophilic protein up to 400 K (the exception to this is the behavior of HBps), indicating that the mutation possibly causes a decrease in the melting point (Tm) of the thermophilic protein. Additionally, the net charge in the mutant protein was calculated to be −2 at 298 K, while the thermophilic (wild-type) and mesophilic proteins have net charges equal to −1 and −4, respectively. Therefore, the mutation increases the charge repulsion potential, destabilizing the structure.

It has not been possible to find experimental data on the formation of this Glu3–Lys62–Glu36 triad or on the site-directed point mutation Lys62Ala in the *Bst*HPr protein. However, it has been reported that the HPr protein from the thermophilic organism *Thermoanaerobacter tengcongensis* (*Tt*HPr) forms the Glu3–Lys62 salt bridge, which might compact the structure between β1 and β4 strands, contributing to its thermal stability. Therefore, the site-directed Lys62Thr was mutated in the *Tt*HPr variant. The authors concluded that the measurements by CD spectra provide firm evidence that the Glu3–Lys62 salt bridge is key in the thermostability of the *Tt*HPr protein [16].

In summary, the Glu3–Lys62–Glu36 triad, formed by the Glu3–Lys62 and Glu36–Lys62 salt bridges, plays a crucial role in the stability of the hydrophobic core of the thermophilic *Bst*HPr protein, avoiding its exposure to the solvent and thereby preventing the breaking of buried hydrogen bonds and decreasing the electrostatic surface potential. Thus, this triad functions as a “natural molecular staple” that keeps the structure packaged and stable against energetic changes caused by the thermal increase.

Finally, the OS/CS ratio as a function of temperature may explain the functional mechanism of the HPr proteins regarding their structural stability. Although these proteins have a common folding (open-faced β-sandwich type), their geometry in the vicinity of the active site has marked differences. For example, the crystallographic structure of *Bs*HPr exhibits a distance of *d*_Nδ1-N_ = 3.9 Å, adopting the CS state, while the *Bst*HPr crystal exhibits the OS state, with a distance equal to *d*_Nδ1-N_ = 8.8 Å [19]. Consequently, these differences complicate the comparison between two different proteins, even when performing MD simulations. These structural changes should directly impact the catalytic activity of HPr proteins. However, direct measurement of activity requires additional MD protocols from those used in this investigation.

On the other hand, the only difference between *Bst*HPr and *Bst*HPrm is the Lys62Ala point mutation; thus, a direct comparison can be made between these proteins. Figure 9 shows that the ratio of OS/CS approaches 1 at low temperatures. However, this equilibrium is considerably affected due to unfolding effects at elevated temperatures. There is a correlation between the structural changes in the region of the Lys62 residue and the behavior of the active center, since the mutant protein exhibits more OS than that of the wild-type one at 400 K. Obviously, both proteins reach OS/CS equilibrium at 450 K, as they are fully unfolded.

## 4. Materials and Methods

### 4.1. Molecular Models

The crystal structures of the HPr proteins, originating from the thermophilic microorganism *B. stearothermophilus* (*Bst*) and the mesophilic microorganism *B. subtilis* (*Bs*), were obtained from the Protein Data Bank (PDB, www.rcsb.org (accessed on 15 March 2024)) with PDB-id entries 1Y4Y [14] and 2HPR, respectively [23]. These structures, known as *Bst*HPr and *Bs*HPr, form an open-faced β-sandwich type with four antiparallel β-strands and three α-helices, resulting in a β_1_α_1_β_2_β_3_α_2_β_4_α_3_ arrangement (Figure 10) [14,16,23]. The *Bst*HPr crystallographic structure was protonated using the PropKa program [24] to achieve neutral pH, and the point mutation Lys62Ala was created using PyMOL version 2.5.2 (https://pymol.org (accessed on 24 March 2024)). To confirm that this mutation is a destabilizing site in the *Bst*HPr protein, we carried out virtual mutations using 6 different methods [25]. Details and comments are presented in Appendix A.

### 4.2. Molecular Dynamic Simulations

GROMACS v2020.3 program [26,27] and the AMBER99SB force field [28] were used to perform the MD simulations. Wild-type *Bst*HPr and mutant *Bst*HPrm proteins were solvated using the SPCE water model [29] inside a dodecahedral box with periodic boundary conditions. The distance between the proteins and the edge of the box was equal to 1.0 nm. The systems’ total charge was neutralized at pH = 7.0 by adding sufficient counterions employing the “genion” function of GROMACS [26]. The systems *Bst*HPr and *Bst*HPrm consisted of 1290 and 1278 protein atoms, 15,084 and 13,131 solvent atoms, 1 and 0 Na^+^ ions, and 2 and 2 Cl^–^ ions, yielding a total of 16,377 and 14,411 atoms, respectively. 

The particle mesh Ewald (PME) algorithm was used to evaluate the long-range electrostatic interactions [30]. Specifically, one Coulomb cutoff equal to 1.0 nm and Fourier spacing of 0.16 nm with a 4th-order cubic interpolation were used. In addition, one cutoff of 1.0 nm for the van der Waals interactions was set. The bond lengths were constrained using the LINCS method [31] with a 2 ps integration step. The “steepest descent” function [32] was used to minimize the energy for both proteins, and the processes converged at 624 and 446 steps (−2.68 × 10^5^ and −2.32 × 10^5^ kJ/mol) for the *Bst*HPr and *Bst*HPrm proteins, respectively. The velocity-rescaling algorithm was used to equilibrate the NVT ensemble [33]; the temperatures were fixed at 298, 333, 362, 400, and 450 K. The systems were simulated at temperatures equal to 400 K and 450 K to accelerate global and atomic fluctuations of structures to activate protein unfolding, since the experimental and simulated timescales for this process are very different [20]. The NPT ensemble was equilibrated at 1.0 bar using the Parrinello–Rahman barostat [34]. Both equilibrations were achieved during 100 ps.

In this work, we performed fifteen simulations of the mutant *Bst*HPrm protein, which were then compared with the simulations obtained of the wild-type *Bst*HPr and mesophilic *Bs*HPr proteins given in reference [20]. This way, the same time and number of configurations were simulated, i.e., 1 μs of simulation and 10,001 configurations. Moreover, we repeated simulations at each temperature in triplicate.

### 4.3. Simulation Analysis

As stated in previous work [20], here we carried out the following analyses: (a) root mean square deviation, (b) radius of gyration, (c) fraction of native contacts, (d) secondary structure profiles, (e) hydrogen bonds, (f) solvent accessible surface area, (g) cluster ILV, and (h) salt bridges. However, we briefly describe some noteworthy keys of these analyses.

(a)Root mean square deviation: The first structural conformation of the DM simulation is used as the reference structure (*t* = 0 ns).(b)Radius of gyration: This parameter is calculated from the protein center of mass.(c)Fraction of native contacts: This indicator is determined using the Best–Hummer–Eaton model [35]. For this calculation, the first conformation of the simulations is defined as the native structure (*t* = 0 ns). The total number of contacts in the native structure is taken as Q = 1, and from this reference, the contacts for the remaining conformations of the trajectory are obtained.(d)Secondary structure profiles: SS assignment is performed using the define secondary structure of proteins (DSSP) algorithm [36]. This algorithm considers 8 types of SS: α-helix, π-helix, 3_10_-helix, β-strand, β-bridge, random coil, bend, and turn. After this calculation, Micsonai et al. classified these structures in three different groups [37], that is, in the α-helix SS the 3 helix structures (α-helix, π-helix, and 3_10_-helix) are included, in the β-strand SS only the β-strand is considered, and in the random coil the remaining structures (β-bridge, random coil, bend, and turn) are included. Micsonai et al. proposed this classification from protein structure data and their respective circular dichroism spectra. Therefore, the set of the three classifications (α-helix, β-strand, and random coil) is considered 100% of the secondary structure.(e)Hydrogen bonds: For this calculation, the distance *r* and the angle θ between the mass centers of the acceptor (A) and donor (D) atoms of the proton (H) are considered (*r*_AD_ ≤ 3.5 Å and θ_AD_ ≤ 30°).(f)Solvent accessible surface area: This parameter is determined using the Lee and Richards approximation: one solvent sphere with a radius of 1.4 Å is used [38].(g)Cluster ILV: The contacts of structural units (CSU) algorithm is used to find the groupings of isoleucine, leucine, and valine residues within proteins [39]. This methodology analyzes atoms as spheres with a van der Waals radius. The contact of two atoms, A and B, is considered, i.e., a test sphere on the surface of atom A must overlap at least 10 Å with the surface of the sphere of atom B. If this contact occurs between the atoms of the residues Ile, Leu, and Val, they are considered part of a cluster. Therefore, different ILV clusters can be expected in the proteins.(h)Salt bridges: The Barlow and Thornton criterion is taken to measure SB formation [40], i.e., *r*_SB_ ≤ 0.4 nm. In addition, the ionic pairs were calculated using the GetContacts program (https://getcontacts.github.io/ (accessed on 22 February 2024)), taking as a criterion of formation that the average frequency must be equal to 0.3 during the trajectories of the three replicas for each temperature.

## 5. Conclusions

By comparing and analyzing the structural and molecular interaction parameters between both proteins (*Bst*HPr and *Bst*HPrm), the salt bridge network formed with the Glu3–Ala62–Glu36 triad is essential for maintaining the thermal stability of the thermophilic protein, since without these interactions two phenomena take place, namely:(a)Global fluctuations increase, compaction/expansion processes increase, topological native contacts decrease, ordered secondary structures are lost, and disordered structures increase;(b)Buried hydrogen bonds decrease and those formed with the solvent increase, while the non-polar residues are more exposed to the solvent, causing a loss of hydrophobic interactions.

This triad acts as a “natural molecular staple” between secondary structures, contributing meaning to the thermal stability of the *Bst*HPr protein. As the mutant protein lacks this triad, its structure is the most affected up to 400 K, mainly losing the secondary structure β_1_- and β_4_-strands and α_1_- and α_3_-helices. Therefore, the mutation of the Lys62 residue by the Ala62 residue is noteworthy, since the thermal resistance of the thermophilic protein decreases, as has been reported in experimental analysis of the site-direct mutation Lys62Thr in the *Tt*HPr protein [16].

## Figures and Tables

**Figure 1 ijms-25-06316-f001:**
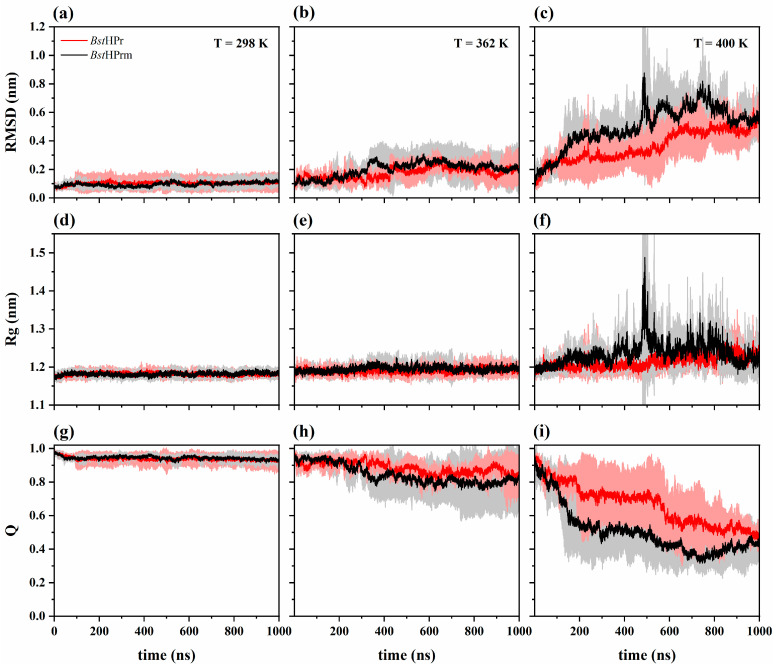
Time evolution of the: (**a**–**c**) RMSD; (**d**–**f**) Rg; and (**g**–**i**) Q for the *Bst*HPr (red line) and *Bst*HPrm (black line) proteins at 298 K, 362 K, and 400 K. The solid lines and shaded areas represent the average values and their standard deviation values from the three replicas of MD simulations, respectively. Details on these trajectories and the average values for the temperatures analyzed are given in Appendix A.

**Figure 2 ijms-25-06316-f002:**
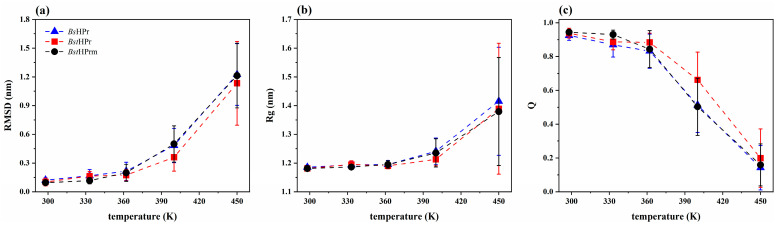
Average values and standard deviations of the (**a**) RMSD, (**b**) Rg, and (**c**) Q for the *Bs*HPr, *Bst*HPr, and *Bst*HPrm proteins at 298, 333, 362, 400, and 450 K. The values have been calculated from the three replicas of MD simulations. Symbols represent the avg values and the SD values are indicated with bars. The dashed lines are only guides for the eye. Appendix A show the corresponding numerical values. Appendix A shows the boxplots for these analyses to provide a more detail examination of the statistical distribution of the data.

**Figure 3 ijms-25-06316-f003:**
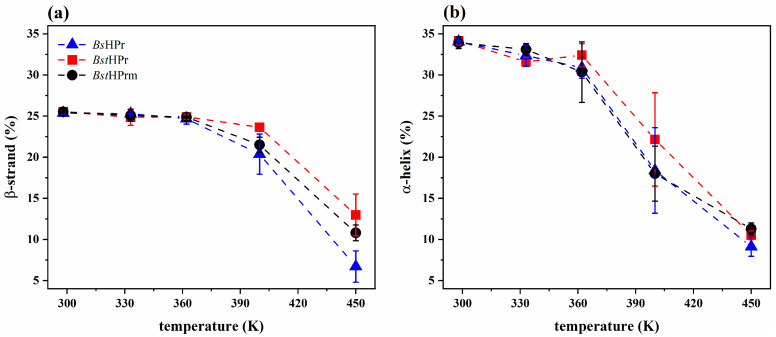
Average percentage values and standard deviations of the native structures: (**a**) β-strands and (**b**) α-helices for the *Bs*HPr, *Bst*HPr, and *Bst*HPrm proteins at 298, 333, 362, 400, and 450 K. The values have been calculated from the three replicas of MD simulations. Symbols represent the avg values and the SD values are indicated with bars. Dashed lines are only guides for the eye. The boxplots of these analyses are given in Appendix A.

**Figure 4 ijms-25-06316-f004:**
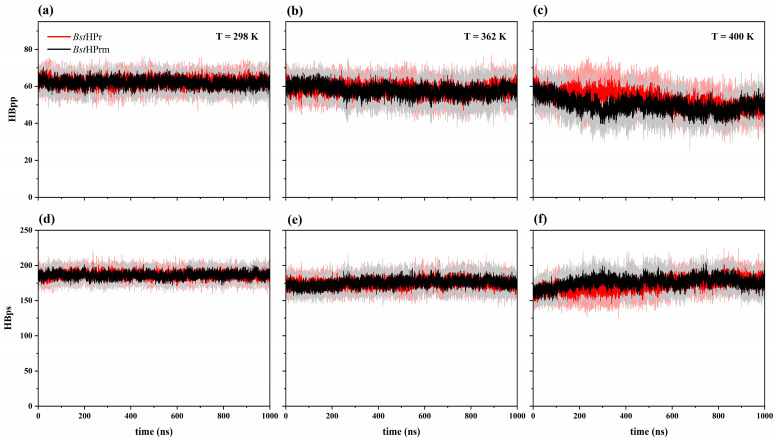
Time evolution of hydrogen bonds for the *Bst*HPr (red line) and *Bst*HPrm (black line) proteins at (**a**) 298 K, (**b**) 362 K, and (**c**) 400 K for HBpp and (**d**) 298 K, (**e**) 362 K, and (**f**) 400 K for HBps. The solid lines and shaded areas represent the average values and their standard deviation values from the three replicas of MD simulations, respectively. Details on these trajectories and the average values for the temperatures analyzed are given in Appendix A.

**Figure 5 ijms-25-06316-f005:**
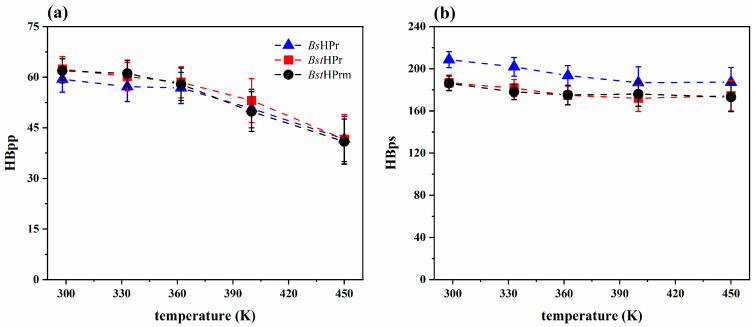
Average values and standard deviation of the (**a**) HBpp and (**b**) HBps for the *Bs*HPr, *Bst*HPr, and *Bst*HPrm proteins at 298, 333, 362, 400, and 450 K. The values have been calculated from the three replicas of the MD simulations. Symbols represent the avg values, and the SD values are indicated with bars. The dashed lines are only guides for the eye. Appendix A shows the corresponding numerical values. The boxplots of these analyses are given in Appendix A.

**Figure 6 ijms-25-06316-f006:**
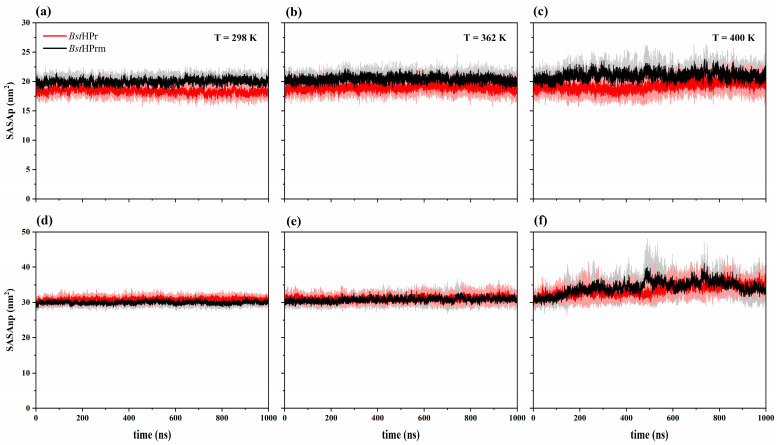
Time evolution of solvent accessible surface area for the *Bst*HPr (red line) and *Bst*HPrm (black line) proteins at (**a**) 298 K, (**b**) 362 K, and (**c**) 400 K for SASAp and (**d**) 298 K, (**e**) 362 K, and (**f**) 400 K for SASAnp. The solid lines and shaded areas represent the average values and their standard deviation from the three replicas of MD simulations, respectively. Details on these trajectories and the average values for the temperatures analyzed are given in Appendix A.

**Figure 7 ijms-25-06316-f007:**
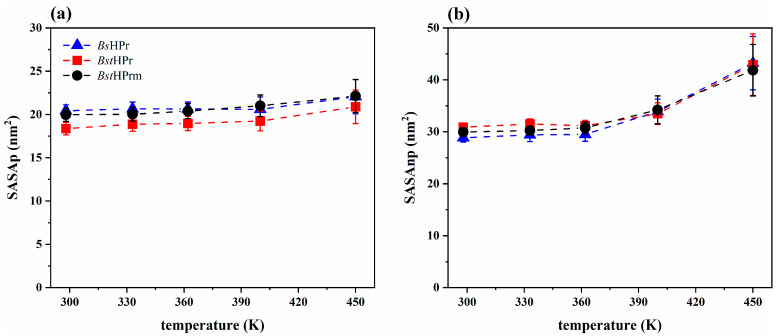
Average values and standard deviation of the (**a**) SASAp and (**b**) SASAnp for the *Bs*HPr, *Bst*HPr, and *Bst*HPrm proteins at 298, 333, 362, 400, and 450 K. The values have been calculated from the three replicas of the MD simulations. Symbols represent the avg values, and the SD values are indicated with bars. The dashed lines are only guides for the eye. Appendix A shows the corresponding numerical values. The boxplots of these analyses are given in Appendix A.

**Figure 8 ijms-25-06316-f008:**
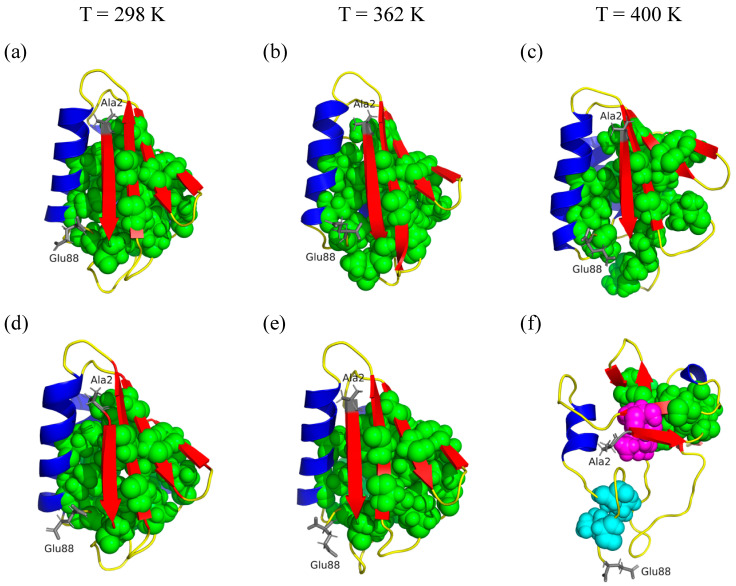
ILV clusters for: (**a**–**c**) *Bst*HPr and (**d**–**f**) *Bst*HPrm proteins at 298, 362, and 400 K. β-strands, α-helices, and random coils are colored in red, blue, and yellow, respectively. ILV clusters were calculated using the ProteinTools tool (https://proteintools.uni-bayreuth.de/ (accessed on 24 March 2024)) [21] and are shown in green, cyan, and magenta. Each snapshot was made using PyMOL version 2.5.2 software for simulation 1 at 750 ns. The initial and final residues of both proteins are shown in gray. Details of these clusters can be found in Appendix A.

**Figure 9 ijms-25-06316-f009:**
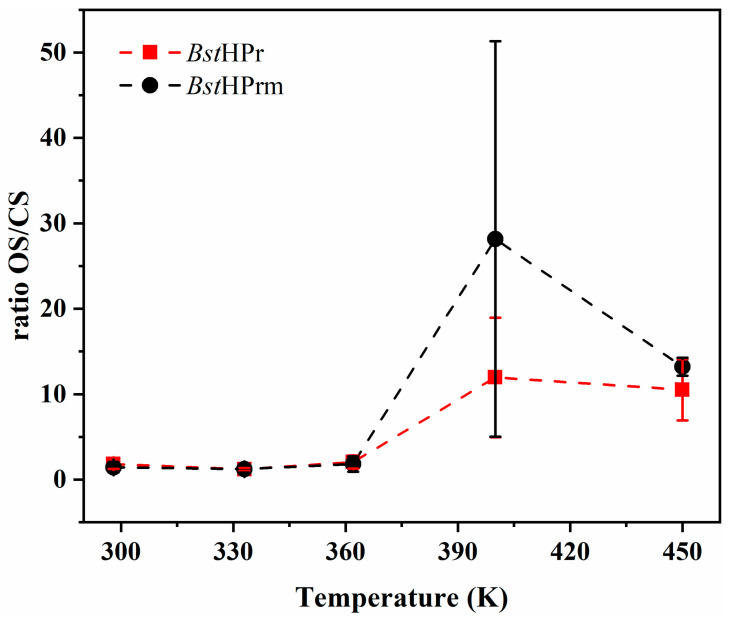
Ratio of OS/CS for *Bst*HPr and *Bst*HPrm proteins at different temperatures of simulation. Symbols represent the avg values, and the SD values are indicated with bars. The dashed lines are only guides for the eye.

**Figure 10 ijms-25-06316-f010:**
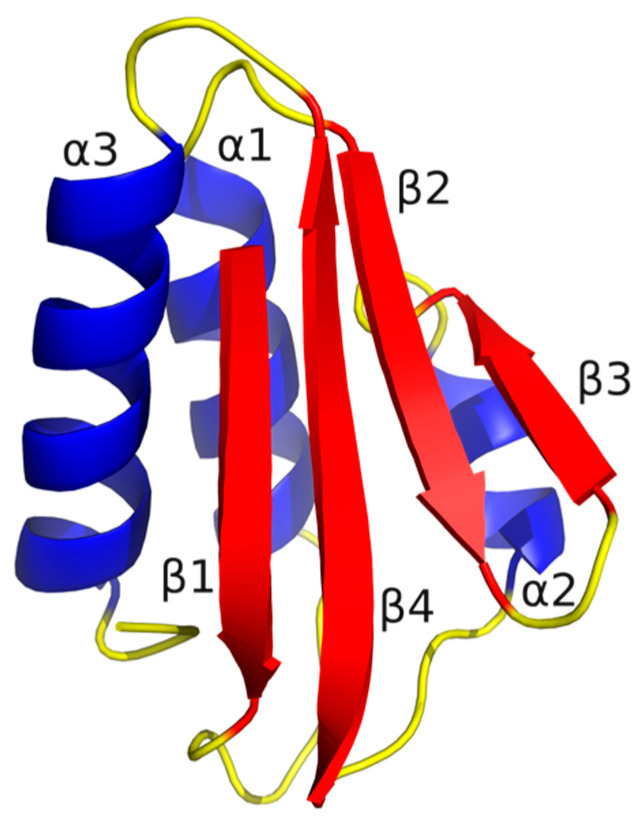
Snapshot of the *Bst*HPr protein at *t* = 0 ns of simulation at 298 K. Structure created using PyMOL version 2.5.2 software. The color codes of the SS are the same as in Figure 8.

**Table 1 ijms-25-06316-t001:** Thermodynamic data of the *Bst*HPr and *Bs*HPr proteins.

Protein	T_S_ (°C)	T_m_ (°C)	∆G_S_ (kcal/mol)	∆C_p_ (kcal/mol K)	∆H (kcal/mol)
*Bst*HPr	24.8	88.9	8.2	1.37	98.6
*Bs*HPr	24.1	74.4	5.2	1.33	76.7

**Table 2 ijms-25-06316-t002:** Approximate number and percentage of HBpp lost for *Bst*HPr and *Bst*HPrm.

Temperature Range	*Bst*HPr		*Bst*HPrm	
	Number	Percentage	Number	Percentage
298–333	2	3.5	1	1.3
298–362	4	6.2	4	6.5
298–400	9	14.9	12	19.5
298–450	21	33.3	21	33.9

**Table 3 ijms-25-06316-t003:** Approximate number and percentage of HBps lost for *Bst*HPr and *Bst*HPrm.

Temperature Range	*Bst*HPr		*Bst*HPrm	
	Number	Percentage	Number	Percentage
298–333	5	2.7	8	4.3
298–362	12	6.4	11	5.9
298–400	15	7.9	10	5.4
298–450	12	6.6	13	7.0

**Table 4 ijms-25-06316-t004:** Increased and decreased area in the *Bst*HPr protein by the mutation.

Temperature	SASAp		SASAnp	
	Area (nm^2^)	Percentage	Area (nm^2^)	Percentage
298	1.59	8.7	0.93	3.0
333	1.15	6.1	1.26	4.0
362	1.42	7.5	0.42	1.3
400	1.77	9.2	0.77 *	2.3 *
450	1.24	5.9	1.08	2.5

* These values increase.

**Table 5 ijms-25-06316-t005:** Average frequencies of salt bridges for the *Bst*HPr protein (table taken from Ref. [20]).

Residue Pairs	Temperature (K)				
	298	333	362	400	450
Asp79–Lys83	0.822	0.773	0.773	0.527	0.341
Glu84–Arg17	0.579	0.495	0.526	0.292	0.104
Asp11–Lys57	0.412	0.397	0.396	0.220	0.081
Glu3–Lys62	0.620	0.617	0.633	0.597	0.088
Glu36–Lys62	0.380	0.403	0.526	0.553	0.203

**Table 6 ijms-25-06316-t006:** Average frequencies of salt bridges for the *Bst*HPrm protein.

Residue Pairs	Temperature (K)				
	298	333	362	400	450
Asp79–Lys83	0.810	0.804	0.733	0.526	0.347
Glu84–Arg17	0.422	0.476	0.406	0.245	0.055
Asp11–Lys57	0.474	0.436	0.336	0.057	0.055
Glu32–Lys45	0.246	0.244	0.334	0.351	0.149

## Data Availability

Data is contained within the article and Appendix A.

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
