# Peer review of "Effect of the Lys62Ala Mutation on the Thermal Stability of BstHPr Protein by Molecular Dynamics"

_ijms, 2024, doi:10.3390/ijms25126316_

Round 1
Reviewer 1 Report
Comments and Suggestions for Authors
The manuscript under consideration presents an investigation into the thermal stability of the BstHPr protein via molecular dynamics simulations, focusing on the impact of the site-directed point mutation Lys62. The study compares the behavior of the mutant thermophilic BstHPrm protein with that of the wild-type thermophilic BstHPr protein and the mesophilic BsHPr protein at five different temperatures. The analysis concludes that as temperature increases, protein stability decreases, with the mutant protein exhibiting greater instability at 400 K compared to the wild-type protein.
While several significant shortcomings undermine its validity and impact, rendering it unsuitable for publication in its current form. Some major issues are listed as follow:
(1)The study relies solely on molecular dynamics simulations to draw conclusions about protein stability. While computational approaches are valuable tools in structural biology, they must be validated by experimental data to ensure their reliability.
(2)The author extensively discusses the significance of thermal stability in the first three paragraphs of the Introduction. However, as a research article, the author should focus more on the progress of studying the thermal stability of the BstHPr protein itself, which is lacking in this section.
(3)Does the temperature simulation at 450K have significance?
(4)The manuscript only simulated and analysed the overall structural changes and the region around Lys62, without considering the correlation with enzyme activity.
(5)Table 3 is not typically used to present the results of molecular dynamics studies.
(6)The conclusion section should be concise. The author has included excessive content in the this section, while lacking experimental data to support the assertions.
Author Response
Reviewer #1
The manuscript under consideration presents an investigation into the thermal stability of the BstHPr protein via molecular dynamics simulations, focusing on the impact of the site-directed point mutation Lys62. The study compares the behavior of the mutant thermophilic BstHPrm protein with that of the wild-type thermophilic BstHPr protein and the mesophilic BsHPr protein at five different temperatures. The analysis concludes that as temperature increases, protein stability decreases, with the mutant protein exhibiting greater instability at 400 K compared to the wild-type protein.
Authors’ Reply: We would like to thank the reviewer for the insightful and constructive comments on our manuscript. We believe his/her valuable feedback has significantly contributed to improving the clarity of our work. We addressed the reviewer’s concerns to the best of our abilities.
Comments
While several significant shortcomings undermine its validity and impact, rendering it unsuitable for publication in its current form. Some major issues are listed as follow:
- The study relies solely on molecular dynamics simulations to draw conclusions about protein stability. While computational approaches are valuable tools in structural biology, they must be validated by experimental data to ensure their reliability.
HPr is a small and globular protein that lacks disulfide bonds and prosthetic groups, therefore it is considered a model protein for folding investigations. This protein fits a simple two-state reversible model in both thermal and solvent unfolding experiments. There is a considerable number of studies that address this phenomenon using experimental methodologies. These studies have resulted in the detailed thermodynamic characterization of several homologous HPr proteins. This evidence, together with the availability of a large number of high-resolution crystallographic structures, makes this system ideal for theoretical unfolding studies. Although it is expected to generate experimental information for future work, this point is beyond the aim and scope of the present work, which is based on elucidating and explaining the key factors, at the molecular level, that provide thermal stability to the HPr protein, in particular to BstHPr.
However, to comply with the reviewer, we have added the following paragraph to the manuscript to ensure the reliability of this computational study (Pages 15 and 16).
It has not been possible to find experimental data on the formation of this Glu3-Lys62-Glu36 triad or on the site-directed point mutation Lys62Ala in BstHPr protein. However, it has been reported that the HPr protein from the thermophilic organism Thermoanaerobacter tengcongensis (TtHPr) forms the Glu3-Lys62 salt bridge, which might compact the structure between β1 and β4 strands contributing to its thermal stability. Therefore, the site-directed Lys62Thr was mutated in TtHPr variant. The authors concluded that the measurements by CD spectra provided firm evidence that the Glu3-Lys62 salt bridge is key in the thermostability of the TtHPr protein [16].
- The author extensively discusses the significance of thermal stability in the first three paragraphs of the Introduction. However, as a research article, the author should focus more on the progress of studying the thermal stability of the BstHPr protein itself, which is lacking in this section.
We appreciate the reviewer’s comment. We have added the following paragraphs to the revised manuscript (Page 2)
The thermostability analyses of the BstHPr protein have been reported in various experimental studies [12–16]. Their structural and thermodynamic data have been compared with those of homologous proteins from mesophilic organisms such as B. subtilis (Bs), E. coli (Ec), E. faecalis (Ef), and M. capricolum (Mc); moderate thermophiles such S. thermophilus (St); thermophiles as T. tengcongensis (Tt); and haloalkaliphilic organisms such as B. halodurans (Bh) and O. iheyensis (Oi) [12,14,16]. Stability measurements of the Bst, Bh, St, Bs, and Oi HPr proteins carried out by thermal and solvent denaturation, under different experimental conditions of pH, salinity, and temperature, have shown that the BstHPr has the highest thermal stability. For example, its free energy of stabilization is ∆GS = 8.2 kcal/mol, the temperature of maximal stability is TS = 27 °C, its melting temperature is Tm = 88.9 °C, and the change of enthalpy is ∆H = 98.6 kcal/mol [12,14]. In addition, it has been observed that the BstHPr protein forms a salt bridge between the Asp11 and Lys57 residues [12,16]. The site-directed Lys57Thr mutation shows that the Asp11-Lys57 salt bridge plays an important role in the thermostability of the BstHPr protein [16]. These facts have shed light on the structural behavior and the mechanisms involved in the thermal stability of the BstHPr protein, but have not been conclusive, since the key molecular interactions that stabilize it at elevated temperatures are still not well understood.
- Does the temperature simulation at 450K have significance?
Although all the simulations were run for 1 ms, it was not possible to fully unfold the proteins above their Tm, therefore the simulations should be run for longer times to observe such unfolding. As the computational time can be very large when doing this, the strategy used is to raise the temperature to observe the unfolding of the proteins, as suggested by Daggett and collaborators “In order to unfold a protein, we perform the simulations at very high temperatures, typically 498 K, or 225 °C. Such drastic measures have been necessary because of the large difference in the experimental timescale for unfolding and that achievable with available computer power.”
(https://doi.org/10.1016/S0022-2836(02)00672-1)
We have added the following sentences to the revised manuscript (Page 16).
The systems simulated at temperatures equal to 400 K and 450 K were carried out to accelerate global and atomic fluctuations of structures, to activate protein unfolding, since the experimental and simulated timescales for this process are very different [20].
- The manuscript only simulated and analysed the overall structural changes and the region around Lys62, without considering the correlation with enzyme activity.
We have added the following paragraph to the revised manuscript (Pages 11 and 12).
Lastly, we performed a brief analysis of the flexibility of the active site in both proteins. The main function of the HPr proteins is to transfer a phosphate group within the PTS system in bacteria [12,16,19]. The flexibility of two residues belonging to the active center of the HPr protein, His15 and Arg17, is essential for this transport. Depending on the spatial relationship between these residues, the HPr protein can adopt two main conformations: open state (OS) and closed state (CS). The OS is considered when the distance between the Nd1 of His15 and N of Arg17 is dNd1-N > 7.5 Å, and the CS when this distance is dNd1-N < 4.5 Å. It has been proposed that dynamic conversion between the OS and the CS is essential for the phosphorylation of this protein and its effect as a transcriptional regulator [19]. Through the simulations performed in this study, it was possible to observe that BstHPr and BstHPrm proteins adopt both states frequently during the MD trajectory, hence the ratio OS/CS was calculated according to the total number of states achieved, as indicated in Figure 9.
Moreover, a new Figure has been added after this paragraph.
In the discussion section, the following paragraphs were added (Page 15).
Finally, the OS/CS ratio as a function of temperature may explain the functional mechanism of the HPr proteins regarding their structural stability. Although these proteins have a common folding (open-faced β-sandwich-type), their geometry in the vicinity of the active site has marked differences. For example, the crystallographic structure of BsHPr exhibits a distance of dNd1-N = 3.9 Å, adopting the CS state, while the BstHPr crystal exhibits the OS state, with a distance equal to dNd1-N = 8.8 Å [19]. Consequently, these differences complicate the comparison between two different proteins, even when performing MD simulations. These structural changes should directly impact the catalytic activity of HPr proteins. However, direct measurement of activity requires additional MD protocols from those used in this investigation.
On the other hand, the only difference between BstHPr and BstHPrm is the Lys62Ala point mutation, thus a direct comparison can be made between these proteins. Figure 9 shows that the ratio OS/CS approaches 1 at low temperatures. However, this equilibrium is considerably affected due to unfolding effects at elevated temperatures. There is a correlation between the structural changes in the region of the Lys62 residue and the behavior of the active center, since the mutant protein exhibits more OS than that of the wild-type one at 400 K. Obviously, both proteins reach OS/CS equilibrium at 450 K, as they are fully unfolded.
- Table 3 is not typically used to present the results of molecular dynamics studies.
Table 3 has been changed by the following sentences (Page 16)
The systems BstHPr and BstHPrm consisted of 1,290 and 1,278 protein atoms; 15,084 and 13,131 solvent atoms; 1 and 0 Na+ ions; and 2 and 2 Cl– ions, giving a total of 16,377 and 14,411 atoms, respectively.
- The conclusion section should be concise. The author has included excessive content in the this section, while lacking experimental data to support the assertions.
The conclusions are now concise. The experimental data given in comment 1 have been added.
Reviewer 2 Report
Comments and Suggestions for Authors
The authors of the manuscript 'Effect of the Lys62Ala Mutation on the Thermal Stability BstHPr Protein by Molecular Dynamics' describe the effect of the Lys62 to Ala62 mutation on the thermal stability of The HPr protein from Bacillus stearothermophilus and compare it to the wildtype version and that of the homolog protein from B. subtilis.
They use the results from triplicate 1 us MD-runs at various temperatures and conclude that the mutation lowers the thermal stability.
Comments:
Salt bridges and charge interactions contribute quite substantially to stability of proteins. It is therefor not surprising that upon removal of such an interaction, the stability of the protein gets reduced.
The reason why this mutation was chosen, and not another saltbridge should be explained.
1) The manuscript is interesting, but quite long and a little repetitive. The conclusions should be shortened.
2) It would be good to see some biochemical data to support the conclusions. Thermal unfolding followed by CD or fluorescence should be performed in parallel using all three proteins.
3) A figure showing the Glu3-Lys62-Glu36 triad in detail would be helpful, and can be compared to the same (partial) structure of the B. subtilis HPr protein.
Minor remarks:
1) Lys62 to ala62 should be mentioned in the abstract.
2) RNAm should be mRNA
3) English spelling and grammar should be checked.
English spelling and grammar should be checked.
Author Response
Reviewer #2
The authors of the manuscript 'Effect of the Lys62Ala Mutation on the Thermal Stability BstHPr Protein by Molecular Dynamics' describe the effect of the Lys62 to Ala62 mutation on the thermal stability of The HPr protein from Bacillus stearothermophilus and compare it to the wildtype version and that of the homolog protein from B. subtilis. They use the results from triplicate 1 us MD-runs at various temperatures and conclude that the mutation lowers the thermal stability.
Authors’ Reply: We would like to thank the reviewer for the insightful and constructive comments on our manuscript. We believe his/her valuable feedback has significantly contributed to improving the clarity of our work. We addressed the reviewer’s concerns to the best of our abilities.
Comments:
- Salt bridges and charge interactions contribute quite substantially to stability of proteins. It is therefor not surprising that upon removal of such an interaction, the stability of the protein gets reduced.
The reason why this mutation was chosen, and not another saltbridge should be explained.
In a previous work (doi.org/10.3390/ijms24119557), we analyzed the thermal stability of the thermophilic protein BstHPr. The formation of the Glu3-Lys62 and Glu36-Lys62 salt bridges was observed in such investigation. It was found that both ion pairs formed from the salt bridge network of the Glu3-Lys62-Glu36 triad. This triad was not observed in the mesophilic BsHPr protein. This motivated us to perform the analysis of the point mutation K62A. Therefore, this mutation disrupts the formation of the triad. It has not been possible to find experimental data on the formation of this triad or about the K62A mutant in BstHPr protein. However, it has been reported that the HPr protein from the thermophilic organism T. tengcongensis (TtHPr) forms the Glu3-Lys62 salt bridge, which might tighten the structure between β1 and β4 strands contributing to its thermostability. With these considerations, the site-directed K62T was mutated in TtHPr. The authors concluded that the measurements by CD spectra provided firm evidence that the Glu3-Lys62 salt bridge is key and important in the thermostability of the TtHPr protein.
Taking these facts into account, we have added the following paragraphs to the manuscript (Pages 15 and 16).
It has not been possible to find experimental data on the formation of this Glu3-Lys62-Glu36 triad or on the site-directed point mutation Lys62Ala in BstHPr protein. However, it has been reported that the HPr protein from the thermophilic organism Thermoanaerobacter tengcongensis (TtHPr) forms the Glu3-Lys62 salt bridge, which might compact the structure between β1 and β4 strands contributing to its thermal stability. Therefore, the site-directed Lys62Thr was mutated in TtHPr variant. The authors concluded that the measurements by CD spectra provided firm evidence that the Glu3-Lys62 salt bridge is key in the thermostability of the TtHPr protein [16].
- The manuscript is interesting, but quite long and a little repetitive. The conclusions should be shortened.
The conclusions are now concise.
- It would be good to see some biochemical data to support the conclusions. Thermal unfolding followed by CD or fluorescence should be performed in parallel using all three proteins.
Thermal unfolding followed by CD or fluorescence experiments is beyond the aim and scope of this work, which is based on elucidating and explaining the key factors, at the molecular level, that provide thermal stability to the HPr protein, in particular to BstHPr. Although it is considered to generate experimental information for future works.
The experimental data given in comment 1 have been added to the conclusions.
- A figure showing the Glu3-Lys62-Glu36 triad in detail would be helpful, and can be compared to the same (partial) structure of the B. subtilis HPr protein.
Figure S14 of the Supplementary Materials shows this recommendation. Moreover, we have added the following sentences to the revised manuscript (Page 11).
The structural comparison between the residues of the Glu3-Lys62-Glu36 triad in BstHPr and the corresponding residues of the BstHPrm and BsHPr proteins are given in Figure S14 of the Supplementary Materials.
- Minor remarks:
- Lys62 to ala62 should be mentioned in the abstract.
- RNAm should be mRNA
- English spelling and grammar should be checked.
These observations have been corrected.

Reviewer 3 Report
Comments and Suggestions for Authors
The investigation examined the impact of the Lys62Ala mutation on the thermal stability of the BstHPr protein through molecular dynamics simulations (MDs). Wild-type BstHPr and mutant BstHPrm proteins were analyzed at temperatures ranging from 298 K to 450 K. Structural analyses revealed that both proteins lost their native secondary structures at higher temperatures, with the wild-type variant exhibiting greater structural stability throughout the simulations.
The article is meticulously written, providing comprehensive reporting of numerous thermodynamic and structural analyses conducted on the protein variants. These analyses encompassed a wide range of parameters, including root mean square deviation (RMSD), radius of gyration (Rg), fraction of native contacts, hydrogen bonds, solvent accessible surface area (SASA), and salt bridges. Additionally, the study compared the behavior of the investigated protein variants with that of a Bacillus subtilis mesophilic protein (BsHPr), offering valuable insights into their thermal stability and structural dynamics across different temperature ranges.
Minor comments:
While the three-dimensional structure of BstHPr is clearly sourced from the Protein Data Bank with the specific identifier provided, the origin of the BsHPr structure remains unspecified in the text. Clarifying the source of the BsHPr structure directly in the article would greatly assist readers in understanding the basis of the comparative analysis. This could be achieved by explicitly stating the origin of the BsHPr structure within the main body of the text, eliminating the need for readers to refer to external references ([18]) for this information.
Similarly, providing clarity on the role of BsHPr earlier in the article would enhance reader understanding. This could be achieved by explicitly stating its significance either at the end of the introduction or within the methodology section. For instance, a sentence could be added at the end of the introduction to highlight that the study also includes comparisons with BsHPr to elucidate differences between thermophilic and mesophilic proteins. Alternatively, within the methodology section, a brief explanation could be given regarding how BsHPr is utilized as a comparative reference for the analysis of BstHPr and BstHPrm. This proactive clarification would ensure that readers are informed of BsHPr's role from the outset, facilitating a better understanding of the study's design and objectives. Since the role of BsHPr in this study is not made explicit until line 104.
Could you provide a more detailed explanation of the methodology used to analyze native contacts and ILV clusters? Regarding the native contacts: It is not clear how authors quantified the % of alpha-helix and beta-strands in Fig. 3, since they start from approximately 35% and 25% at 300 K, respectively.
I recommend presenting the information detailed in lines 190-199 and 238-243 in a tabular format. This approach would enhance the clarity of your analysis and assist readers in comprehending the key findings more effectively.
I suggest accompanying the analysis of simulation time with a boxplot to provide a more detailed examination of the statistical distribution of the data. Additionally, incorporating statistical tests such as the Student's t-test or Mann-Whitney U test would further enhance the rigor of the analysis. While line plots effectively illustrate the temporal evolution of thermodynamic or structural variables, they may lack granularity in depicting specific differences due to their reliance on average and standard deviation values. Utilizing boxplots and statistical tests would offer a more comprehensive understanding of the data and strengthen the interpretation of results.
Author Response
Reviewer #3
The investigation examined the impact of the Lys62Ala mutation on the thermal stability of the BstHPr protein through molecular dynamics simulations (MDs). Wild-type BstHPr and mutant BstHPrm proteins were analyzed at temperatures ranging from 298 K to 450 K. Structural analyses revealed that both proteins lost their native secondary structures at higher temperatures, with the wild-type variant exhibiting greater structural stability throughout the simulations.
The article is meticulously written, providing comprehensive reporting of numerous thermodynamic and structural analyses conducted on the protein variants. These analyses encompassed a wide range of parameters, including root mean square deviation (RMSD), radius of gyration (Rg), fraction of native contacts, hydrogen bonds, solvent accessible surface area (SASA), and salt bridges. Additionally, the study compared the behavior of the investigated protein variants with that of a Bacillus subtilis mesophilic protein (BsHPr), offering valuable insights into their thermal stability and structural dynamics across different temperature ranges.
Authors’ Reply: We would like to thank the reviewer for the insightful and constructive comments on our manuscript. We believe his/her valuable feedback has significantly contributed to improving the clarity of our work. We addressed the reviewer’s concerns to the best of our abilities.
The revised version has been improved, which is highlighted in magenta in the text.
Minor comments:
- While the three-dimensional structure of BstHPr is clearly sourced from the Protein Data Bank with the specific identifier provided, the origin of the BsHPr structure remains unspecified in the text. Clarifying the source of the BsHPr structure directly in the article would greatly assist readers in understanding the basis of the comparative analysis. This could be achieved by explicitly stating the origin of the BsHPr structure within the main body of the text, eliminating the need for readers to refer to external references ([18]) for this information.
We have added the following words to complete some paragraphs to the revised manuscript.
(Page 3)
Moreover, the effects of temperature on the mutant structure of the HPr protein were analyzed and compared with those obtained from the thermophilic wild-type (BstHPr) and mesophilic (BsHPr) structures.
(Page 16)
The crystal structures of the HPr proteins, originating from the thermophilic microorganism B. stearothermophilus (Bst) and mesophilic microorganism B. subtilis (Bs) were obtained from the Protein Data Bank (PDB, www.rcsb.org (accessed on 15 March 2024)) with PDB-id entries 1Y4Y [14] and 2HPR, respectively [23]. These structures, known as BstHPr and BsHPr,
- Similarly, providing clarity on the role of BsHPr earlier in the article would enhance reader understanding. This could be achieved by explicitly stating its significance either at the end of the introduction or within the methodology section. For instance, a sentence could be added at the end of the introduction to highlight that the study also includes comparisons with BsHPr to elucidate differences between thermophilic and mesophilic proteins. Alternatively, within the methodology section, a brief explanation could be given regarding how BsHPr is utilized as a comparative reference for the analysis of BstHPr and BstHPrm. This proactive clarification would ensure that readers are informed of BsHPr's role from the outset, facilitating a better understanding of the study's design and objectives. Since the role of BsHPr in this study is not made explicit until line 104.
We have added the following table in the introduction section of the revised manuscript (Page 3).
The thermostability analyses of the BstHPr protein have been reported in various experimental studies [12–16]. Their structural and thermodynamic data have been compared with those of homologous proteins from mesophilic organisms such as B. subtilis (Bs), E. coli (Ec), E. faecalis (Ef), and M. capricolum (Mc); moderate thermophiles such S. thermophilus (St); thermophiles as T. tengcongensis (Tt); and haloalkaliphilic organisms such as B. halodurans (Bh) and O. iheyensis (Oi) [12,14,16]. Stability measurements of the Bst, Bh, St, Bs, and Oi HPr proteins carried out by thermal and solvent denaturation, under different experimental conditions of pH, salinity, and temperature, have shown that the BstHPr has the highest thermal stability. For example, Table 1 shows a comparison of thermodynamic data, such as free energy of stabilization (∆GS), temperature of maximal stability (TS), melting temperature (Tm), change in heat capacity (∆Cp), and change in enthalpy (∆H), between the BstHPr and BsHPr proteins at pH = 7.0 [12–14].
Table 1. Thermodynamic data of the BstHPr and BsHPr proteins.
|
Protein |
TS (°C) |
Tm (°C) |
∆GS (kcal/mol) |
∆Cp (kcal/mol K) |
∆H (kcal/mol) |
|
BstHPr |
24.8 |
88.9 |
8.2 |
1.37 |
98.6 |
|
BsHPr |
24.1 |
74.4 |
5.2 |
1.33 |
76.7 |
In addition, it has been observed that the BstHPr protein forms a salt bridge between the Asp11 and Lys57 residues [12,16]. The site-directed Lys57Thr mutation shows that the Asp11-Lys57 salt bridge plays an important role in the thermostability of the BstHPr protein [16]. These facts have shed light on the structural behavior and the mechanisms involved in the thermal stability of the BstHPr protein, but have not been conclusive, since the key molecular interactions that stabilize it at elevated temperatures are still not well understood.
- Could you provide a more detailed explanation of the methodology used to analyze native contacts and ILV clusters? Regarding the native contacts: It is not clear how authors quantified the % of alpha-helix and beta-strands in Fig. 3, since they start from approximately 35% and 25% at 300 K, respectively.
We have added the following sentences to the revised manuscript (Page 5).
Figure 3 shows the behavior of the β-strand and α-helix native secondary structures (SS) of the three proteins. These proteins contain the same amount of secondary structures at 300 K, i.e., the content of β-strand and α-helix is approximately 26 and 34%, respectively, thus around 40% of the secondary structure is random coil, which is not shown here. Moreover, it is observed that with increasing temperature, the BsHPr and BstHPrm proteins loss β-strand and α-helix structures compared to the BstHPr protein.
We have added the following explanation in the Simulation analysis sub-section of the revised manuscript (Page 17).
- Root mean square deviation: the first structural conformation of the DM simulation is used as the reference structure (t = 0 ns).
- Radius of gyration: this parameter is calculated from protein center of mass.
- Fraction of native contacts: this indicator was determined using the Best-Hummer-Eaton model [34]. For this calculation, the first conformation of the simulations is defined as the native structure (t = 0 ns). The total number of contacts in the native structure is taken as Q = 1 and from this reference, the contacts for the remaining conformations of the trajectory are obtained.
- Secondary structure profiles: SS assignment was performed using the Define Secondary Structure of Proteins (DSSP) algorithm [35]. This algorithm considers 8 types of SS: α-helix, π-helix, 310-helix, β-strand, β-bridge, random coil, bend, and turn. Later to this calculation, Micsonai et al. classified these structures in three different groups [36], that is, in the α-helix SS the 3 helix structures (α-helix, π-helix, and 310-helix) are included, in the β-strand SS only β-strand is considered, and in the random coil the remaining structures (β-bridge, random coil, bend, and turn) are included. Micsonai et al. proposed this classification from protein structure data and their respective circular dichroism spectra. Therefore, the set of the three classifications (α-helix, β-strand, and random coil) is considered as 100% of secondary structure.
- Hydrogen bonds: for this calculation, the distance r and the angle q between the mass centers of the acceptor (A) and donor (D) atoms of the proton (H) are considered (rAD ≤ 3.5 Å and qAD ≤ 30°).
- Solvent accessible surface area: this parameter was determined using the Lee and Richards’s approximation: one solvent sphere with a radius of 1.4 Å is used [37].
- Cluster ILV: the Contacts of Structural Units (CSU) algorithm is used to find the groupings of isoleucine, leucine, and valine residues within proteins [38]. This methodology analyzes atoms as spheres with van der Waals radius. The contact of two atoms A and B is considered, i.e., a test sphere on the surface of atom A must overlap at least 10 Å with the surface of the sphere of atom B. If this contact occurs between the atoms of the residues Ile, Leu, and Val, they are considered part of a cluster. Therefore, different ILV clusters can be expected in the proteins.
- Salt bridges: the Barlow and Thorton criterion has been taken to measure SB formation [39], i.e., rSB ≤ 0.4 nm. In addition, the ionic pairs were calculated using the GetContacts program (https://getcontacts. github.io/ (accessed on 22 February 2024)), taking as a criterion of formation that the average frequency must be equal to 0.3 during the trajectories of the three replicas for each temperature.
Morover, we have added five new references to the revised manuscript (references 34, 35, 36, 37, and 38).
- I recommend presenting the information detailed in lines 190-199 and 238-243 in a tabular format. This approach would enhance the clarity of your analysis and assist readers in comprehending the key findings more effectively.
We have added these observations. The revised manuscript content three new tables (Tables 2, 3, and 4) in Pages 7, 8, and 9.
- I suggest accompanying the analysis of simulation time with a boxplot to provide a more detailed examination of the statistical distribution of the data. Additionally, incorporating statistical tests such as the Student's t-test or Mann-Whitney U test would further enhance the rigor of the analysis. While line plots effectively illustrate the temporal evolution of thermodynamic or structural variables, they may lack granularity in depicting specific differences due to their reliance on average and standard deviation values. Utilizing boxplots and statistical tests would offer a more comprehensive understanding of the data and strengthen the interpretation of results.
We have added for each analysis of simulation time a boxplot to provide a more detail examination. All the boxplots are added to the revised Supplementary Materials, but they are cited in the corresponding figure captions of the revised manuscript.
We have not incorporated the statistical tests such as the Student's t-test or Mann-Whitney U, because we think that the boxplots are sufficient to clarify this comment. But if the reviewer requires these calculations, we will do them in the next round.

Round 2
Reviewer 1 Report
Comments and Suggestions for Authors
The revised manuscript shows improvement with the addition of supplementary content; however, several key issues remain unresolved.
(1) As raised by the reviewer in the first round, there have been many reports on molecular dynamics research of BstHPr protein. The manuscript only conducted classical molecular dynamics analysis at multiple temperature levels and did not combine some other methods, such as replica exchange molecular dynamics;
(2) As raised by the reviewer in the first round, the study primarily relies on molecular dynamics analysis and lacks experimental validation. While the author justifies this focus on theoretical aspects, it is suggested that the impact of key residue pairs on stability, as indicated in the research results, could benefit from further confirmation through sequence analysis or virtual mutation methods.
Author Response
Reviewer #1, Round 2
Authors’ Reply: We thank the reviewer again for sending us their expert notes on our work. We have carefully considered such comments, and have modified and improved our manuscript accordingly.
The revised manuscript shows improvement with the addition of supplementary content; however, several key issues remain unresolved.
Comments
(1) As raised by the reviewer in the first round, there have been many reports on molecular dynamics research of BstHPr protein. The manuscript only conducted classical molecular dynamics analysis at multiple temperature levels and did not combine some other methods, such as replica exchange molecular dynamics;
In our work group, we have not yet implemented alternative molecular simulation techniques such as replica exchange. Although it is highly recommended to use these enhanced sampling methods to explore the conformational space of biomolecules, describing protein folding using these techniques remains an inaccessible problem with current computational tools. A viable alternative is to increase the temperature in the simulations to obtain exhaustive sampling of the N-U pathway. As stated in the first round (point 3), one strategy to unfold proteins using molecular dynamics is to raise the temperature to conditions above the melting temperature (Tm). However, in order to avoid these elevated temperatures, for future work, our research group has considered performing simulations at lower and slightly higher temperatures above the systems’ melting point, significantly increasing the simulation time (5 to 10 ms). We have used other alternatives to analyze protein unfolding and stability, for example, a recently published paper has used the free energy profiles obtained from unfolding simulations at extreme temperatures to yield experimentally compared conclusions (López-Pérez et al., 2023). Although the magnitudes of DG are not comparable with the experimental ones, the contributions of the different conformations of isolated b subunit ATP synthase allowed us to elucidate its stabilization mechanism. Therefore, the strategy reported in that paper is a clarifying alternative to be applied in other systems such as HPr proteins.
López-Pérez, E.; Tuena de Gómez-Puyou, M.; Nuñez, C.J.; Martínez Zapién, D.; Alas Guardado S.; Beltrán, H.I.; Pérez-Hernánez, G. Ordered-domain Unfolding of Thermophilic Isolated β subunit ATP synthase. Protein Science. 2023, 32(7), e4689.
https://doi.org/10.1002/pro.4689
(2) As raised by the reviewer in the first round, the study primarily relies on molecular dynamics analysis and lacks experimental validation. While the author justifies this focus on theoretical aspects, it is suggested that the impact of key residue pairs on stability, as indicated in the research results, could benefit from further confirmation through sequence analysis or virtual mutation methods.
We have added the following sentences to the revised manuscript (Page 16).
To confirm that this mutation is a destabilizing site in the BstHPr protein, we carried out virtual mutations using 6 different methods. Details and comments are presented in Table S8 of the Supplementary Materials.
We have added the following information to the Supplementary Materials (Page 25).
Virtual predictor analyses: predicted DDG values of the mutant protein compared with the wild-type one using 6 virtual predictors.
|
Table S8. Predicted DDG values. |
|
|
Name of predictor |
DDG (kcal/mol) |
|
mCSM1 |
-0.791 |
|
DUET2 |
-0.519 |
|
ENCoM3 |
-0.460* |
|
DynaMut4 |
-0.061 |
|
INPS-MD5 |
-0.595 |
|
MAESTRO6 |
1.621 |
Negative DDG values for mCSM, DUET, ENCoM, DynaMut, and INPS-MD predictors and positive DDG value for MAESTRO predictor [4] indicate that the Lys62Ala mutation is destabilizing, i.e., these virtual predictors show that the BstHPr protein undergoes destabilization from this mutation.
1mCSM: predicting the effects of mutations in proteins using graph-based signatures.
https://biosig.lab.uq.edu.au/mcsm/
2DUET: a server for predicting effects of mutations on protein stability via an integrated computational approach.
https://biosig.lab.uq.edu.au/duet/
3ENCoM: exploring protein conformational space and the effect of mutations on protein function and stability. *This value was calculated using DUET.
https://www.ncbi.nlm.nih.gov/pmc/articles/PMC4489264/
4DynaMut: analysis and prediction of protein stability changes upon mutation using Normal Mode Analysis.
https://biosig.lab.uq.edu.au/dynamut/
5INPS-MD: web server devised to prediction of protein stability change upon single point mutation.
https://inpsmd.biocomp.unibo.it/welcome/default/index
6MAESTRO: predictor based on a multi-agent machine learning system estimation.
https://pbwww.services.came.sbg.ac.at/maestro/web.

Reviewer 2 Report
Comments and Suggestions for Authors
Comments and questions adequatly addressed.
Author Response
We thank the reviewer again.

Round 3
Reviewer 1 Report
Comments and Suggestions for Authors
The author has made significant revisions, and the quality of the paper has been greatly improved. It is recommended to accept.